# Efficient Microwave-Assisted Extraction of Nitrites from Cured Meat and Their Voltammetric Detection at Chemically Modified Electrodes Based on Hexamethyl-p-Terphenyl Poly(methylatedbenzimidazolium) Incorporating Nitrogen-Doped Graphite Nanoplatelets

**Sandra Hernandez-Aldave** [1,†] , **Afshin Tarat** [2] **and Paolo Bertoncello** [1,*]

1    Department of Chemical Engineering, Faculty of Science and Engineering, Bay Campus, Swansea University, Crymlyn Burrows, Swansea SA1 8EN, UK; s.hernandezaldave@swansea.ac.uk or s.hernandezaldave@hud.ac.uk
2    Perpetuus Advanced Materials, Unit B1, Olympus Court, Millstream Way, Swansea Vale, Llansamlet SA7 0AQ, UK; afshintarat@perpetuuscarbon.com
*    Correspondence: p.bertoncello@swansea.ac.uk
†    Current address: Department of Chemical Sciences, School of Applied Sciences, University of Huddersfield, Huddersfield HD1 3DH, UK.

**Abstract:** We describe a fast and reliable procedure for the efficient extraction of nitrites in cured meat using microwave-assisted heat and report their in situ determination via voltammetry using an anion-exchanger ionene, hexamethyl-p-terphenyl poly(benzimidazolium) (HMT-PMBI), and nitrogen-doped graphite nanoplatelets (NGNPs). Cyclic voltammetry and chronoamperometry were utilized to evaluate the concentration of the redox mediator within the film and apparent diffusion coefficient. To investigate the suitability of the composite material for sensing applications, HMT-PMBI/NGNPs were tested for their detection of nitrite in bacon samples without the need of any pretreatments or dilutions. HMT-PMBI/NGNP coated electrodes showed enhanced sensitivity in the detection of nitrite ions in bacon with a limit of detection (LoD) of 0.64 μM, sensitivity 0.52 μA μM$^{-1}$ cm$^{-2}$, and operating in a linear range between 1–300 μM. The results highlight that the determination of nitrites in cured meat using microwave extraction is in good agreement with standard procedures such as the ISO 2918 and the AOAC International 973.31 methods.

**Keywords:** chemically modified electrodes; voltammetry; sensors; nitrites; cured meat; microwave-assisted heat

## 1. Introduction

The use of preservatives in the food industry is of paramount importance in order to maintain the organoleptic properties of food, as well as to protect food from microbial contamination [1,2]. Among the various preservatives used in the food industry, nitrite (NO$_2^-$) ions are among the most widely used, especially in preserving cured meat [3]. For instance, NO$_2^-$ ions are used to inhibit the formation of bacteria such as *Clostridium Botulinum*, which, if present in food, would cause the formation of botulin toxin, which is responsible for neuronal complications and muscular paralysis [4]. NO$_2^-$ ions are also added to meat due to their reaction with myoglobin leading to the preservation of the red color in treated meat. However, high intakes of NO$_2^-$ pose a risk to human health, as nitrites react with secondary biogenic amines within the acidic environment of the stomach, leading to the formation of carcinogenic N-nitroso compounds [5,6]. As a result, several national health bodies have set recommendations for red meat intake, and several countries have introduced legislation to limit the concentration of nitrites added to food. For example, the European Food Safety Authority's guidelines set limitations of nitrites at 0.07 (mg/kg

body weight/day), and the Environmental Protection Agency in the US adopted very similar limits [7]. As such, the quantification of nitrites in processed meat is of relevance to food quality control. There are several analytical methods for the quantification of nitrites, and the most commonly used is the colorimetric (Griess) method developed more than a century ago [8–12]. However, the Griess colorimetric method requires the use of organic compounds for the formation of azo dyes, which is time consuming and results in the formation of toxic byproducts [13,14]. While alternative chromatographic [15–17], spectrophotometric [18–20], and colorimetric [21–24] techniques have recently been developed, electrochemical methods are particularly attractive due to their high sensitivity and selectivity, associated with a fast response time and ease of use [25–32]. More specifically, an anodic determination is preferred to a cathodic one to avoid interferences from nitrate and oxygen reduction [33,34]. Nonetheless, the oxidation of nitrite is characterized by a relatively high overpotential at glassy carbon electrodes, which can be overcome by coating the electrode surface with molecules able to reduce the overpotential and, at the same time, to preconcentrate nitrites. These chemically modified electrodes are very attractive, as they allow detection of analytes at very low concentrations, while protecting the electrode surface from fouling and potential interference. Currently, the most commonly used official reference methods for the extraction and quantitative determination of nitrites in cured meat are (i) the International Organization for Standardization (ISO) 2918 method [35] and (ii) the Association of Official Agricultural Chemists (AOAC International) method 973.31 [36,37]. Both methods are based on spectrophotometry, requiring the use of several reagents as well as lengthy steps to remove the protein content. The use of microwave-assisted heat (MAH) is a well-established procedure widely utilized in analytical chemistry to extract organic and inorganic species from complex matrices [38–44]. Notably, MAH has previously been used to extract nitrosamines from cured meat [38,41]. In this work, we have developed an easy and fast method to extract nitrites from cured meat using only the heat and pressure generated by the microwaves and water as a solvent. Concurrently, we have utilized voltammetric methods to detect nitrites up to submicromolar concentrations using a novel ionene derivative material, namely a HMT-PMBI (hexamethylterphenyl = HMT; poly(methylatedbenzimidazolium) = PMBI))/Nitrogen-doped graphite nanoplatelet (NGNP) composite coated electrode. The determination of nitrite ions achieved using the microwave treatment and the voltammetric method led to results in close agreement with those obtained from the official reference methods, such as the ISO 2918 and the AOAC 973.31 methods.

## 2. Materials and Methods

### 2.1. Reagents and Solutions

Nitrogen-doped graphite nanoplatelets (NGNPs) (surface modified friable nanographite, in the form of graphene nanoflakes, see Figure S1) were obtained from Perpetuus Carbon Technologies Ltd. (UK). NGNPs were synthesized by employing a novel dry physical treatment followed by dielectric barrier discharge plasma with various working gases [45]. HMT-PMBI was synthesized using the procedure developed by Wright et al. [46,47]. The structure (see Figure S2) represents HMT-PMBI in hydroxide form. The HMT-PMBI polymer is composed of three distinct units: the repeat unit A symbolizes a monomer unit possessing 50% degree of methylation (dm), unit B represents 75% dm, and unit C illustrates a unit possessing 100% dm. HMT-PMBI with a dm of 92% was used in this work due to its high ion exchange capacity and its insolubility in water. HMT-PMBI is an anion-exchange polymer and as such it is positively charged. Potassium hexachloroiridate(IV) ($K_2IrCl_6$), NaCl, $NaNO_2$ and sodium tetraborate (99%; $Na_2B_4O_7$) were purchased from Sigma–Aldrich (St. Louis, MO, USA) and used as received. Hexaammineruthenium(III) chloride ($Ru(NH_3)_6)Cl_3$) was purchased from Strem Chemicals (UK). Glassy Carbon electrodes, GCEs, (3 mm diameter), Ag/AgCl reference electrodes, and Pt wires were purchased from IJ Cambria (UK). GCEs were cleaned by successive polishing to gain a mirror-like appearance using 1 μm, 0.3 μm, and 0.05 μm alumina slurry

on micro cloth pads (Buehler), and then washed with methanol. Before use, the GCEs were cycled in 0.1 M sulfuric acid to remove any alumina residue. Aqueous solutions were prepared using Milli-Q Ultra-pure water (with resistivity of 18.2 MΩ cm at 25 °C) from a Millipore Direct Q3 water dispenser (Merck, Watford, UK).

### 2.2. Apparatus

UV-vis spectra were acquired using a double Beam Hitachi U-2900/2910 spectrophotometer. Cyclic voltammetry (CV) measurements were performed using a CH Instrument Model 705 E electrochemical workstation using a conventional three-electrode cell. GCE was used as the working electrode, with a platinum wire as the counter electrode. All potentials were quoted with respect to the Ag/AgCl (3 M) reference electrode, and all the measurements were recorded at room temperature (20 °C). The HMT-PMBI and HMT-PMBI/NGNP coated films were measured using a Taylor Hobson (Leicester, UK), Talysurf stylus profilometer, while pH measurements were taken using a Hanna Instruments (Bedfordshire, UK) 2002 Edge pH meter. The microwave radiation heat used to extract the nitrite from the meat products was produced using a Daewoo KOR6N9RR microwave oven with an 800 W microwave output.

### 2.3. Preparation of HMT-PMBI, and HMT-PMBI/NGNP Composite Solutions

HMT-PMBI/NGNP composite solutions were prepared by the addition of the corresponding concentration of HMT-PMBI into a dispersion of NGNPs in ethanol. For example, a 1 wt% HMT-PMBI and 0.25 wt% NGNP composite solution was prepared by initially dispersing 0.16 g of NGNPs in 40 mL of ethanol, which was sonicated for 30 min. Separately, a 2 wt% HMT-PMBI was prepared by dissolving 85 mg of dry HMT-PMBI membrane in 5 mL ethanol, which was sonicated for 30 min. Finally, a mixture containing 1 mL of each of the two solutions was sonicated for 30 min.

### 2.4. Preparation of HMT-PMBI, and HMT-PMBI/NGNP Coated Electrodes

A 3 mm diameter glassy carbon electrode was used in all electrochemical measurements. The electrode was initially coated by drop casting 2 µL of the PMBI and/or PMBI/NGNP solutions. The composite material film deposited on the electrode surface was dried at room temperature for 10 min.

### 2.5. Extraction of Nitrites Using Microwave-Assisted Heating

The meat product was finely minced and transferred to a Duran glass laboratory bottle with a screw cap. After introducing the appropriate amount of deionized water, the bottle was tightly closed and introduced into the microwave. The extraction of nitrite was based on both the heat and pressure built up inside the bottle during the microwave process, during which the microwaves slowly heated the water and brought it to boiling point. Different parameters were analyzed in order to optimize the microwave extraction method such as (i) the time of the microwave process, (ii) the number of microwave repetitions, and (iii) the amount of bacon used with respect to the volume of water added. First, we assessed the effect of the microwave time on the extraction process. A sample containing 5 g of minced bacon and 200 mL of deionized water in a 250 mL Duran glass laboratory bottle with screw cap was used to find the best conditions according to the following four methods:

**Method 1**. The sample was continuously microwaved until the water warmed up. This occurred after around 120 s.

**Method 2**. The sample was continuously microwaved until the water was visibly boiling inside the glass bottle, which corresponded to 210 s.

**Method 3**. The sample was microwaved for 60 s, then the bottle was shaken and microwaved again for another 60 s. This process was repeated until the water started boiling, which corresponded to ca. 270 s of microwave time.

**Method 4**. The sample was microwaved for 120 s. Then, the sample was left to rest for 600 s. The process was repeated twice more, until the water started boiling; the total microwave time was 360 s.

After completing each of these methods, the various solutions were cooled to room temperature. It is worth mentioning that at a high volume of water (volume > 100 mL) the boiling process was visible for a while even after stopping the microwave process. The cooling process took around 30 min. The resulting solutions were filtered using a fluted filter paper. Finally, adhering to the ISO and AOAC methods (see Supplementary Materials, S3a,b), the solution was diluted and transferred into a 500 mL volumetric flask before being brought to volume. An aliquot of the final solution was taken to react with the color development reagents, and the absorbance value read 540 nm.

## 3. Results and Discussions

### 3.1. Electrochemical Detection of Nitrites

In our previous paper, we demonstrated the effectiveness of HMT-PMBI in preconcentrating negatively charged redox species such as $IrCl_6^{2-}$ [48]. In a similar fashion, using electrochemical characterization we calculated parameters such as the concentration of the redox mediator within the coating and the apparent diffusion coefficient of HMT-PMBI/NGNP coated films (see Supplementary Materials, S4–S9). The preliminary characterization demonstrated that HMT-PMBI and HMT-PMBI/NGNP coated electrodes are suitable to preconcentrate negatively charged redox mediators. For instance, the addition of NGNPs led to a threefold increase in the active surface area of the electrode with a concomitant increase in the apparent diffusion coefficient of more than one order of magnitude compared to HMT-PMBI. In order to ascertain the possibility of using the coated electrodes for the detection of nitrites in cured meat, we first tested the electrochemical behavior of the composite materials in standard solutions containing nitrite ions. Figure 1 shows the linear sweep voltammograms (LSVs) of bare GCE, HMT-PMBI, and HMT-PMBI/NGNP coated electrodes in a solution containing various concentrations of $NO_2^-$ from 1 μM (a) to 100 μM (d).

The LSVs show the typical irreversible oxidation peak of $NO_2^-$ ions to $NO_3^-$ according to the following mechanism [49]:

$$NO_2^- + H_2O \rightarrow NO_3^- + 2H^+ + 2e^- \tag{1}$$

Interestingly, on the bare GCE the oxidation peak was observed to be 0.97 V, while for HMT-PMBI and HMT-PMBI/NGNPs the oxidation of nitrite ions occured at a less positive potential, i.e., 0.95 V and 0.85 V, respectively, as an indication that the oxidation was facilitated using HMT-PMBI and that the addition of NGNPs negatively shifted the oxidation peak potential even further. It is noticeable that when the concentration of $NO_2^-$ was 1 μM, the voltammetric peaks were barely visible for both GCE and HMT-PMBI, whereas a distinct peak was observed for HMT-PMBI/NGNP coated electrodes. This indicates that the addition of NGNPs had a beneficial impact by facilitating the electrochemical oxidation and at the same time enhancing the sensitivity of the modified electrode. As the electrochemical irreversible oxidation of nitrite to nitrate occurs via the formation of two protons and two electrons, we studied the dependence of the voltammetric peak on the pH. The data reported in Figure S10 show that the highest current was observed using pH 7, whereas both at very low (pH 2) and at very high pH values the peak potential shifted ca. 0.15 V towards a more positive potential. This behavior was expected, as at high pH values the concentration of hydroxyl ions is greater, leading to $OH^-$ to complete with $NO_2^-$ for the ion-exchange sites of HMT-PMBI. Instead, at very low pH values, the intensity of the peak decreased as a result of the equilibrium of the reaction as follows [49]:

$$3NO_2^- + 2H^+ \rightarrow NO_3^- + 2NO + H_2O \tag{2}$$

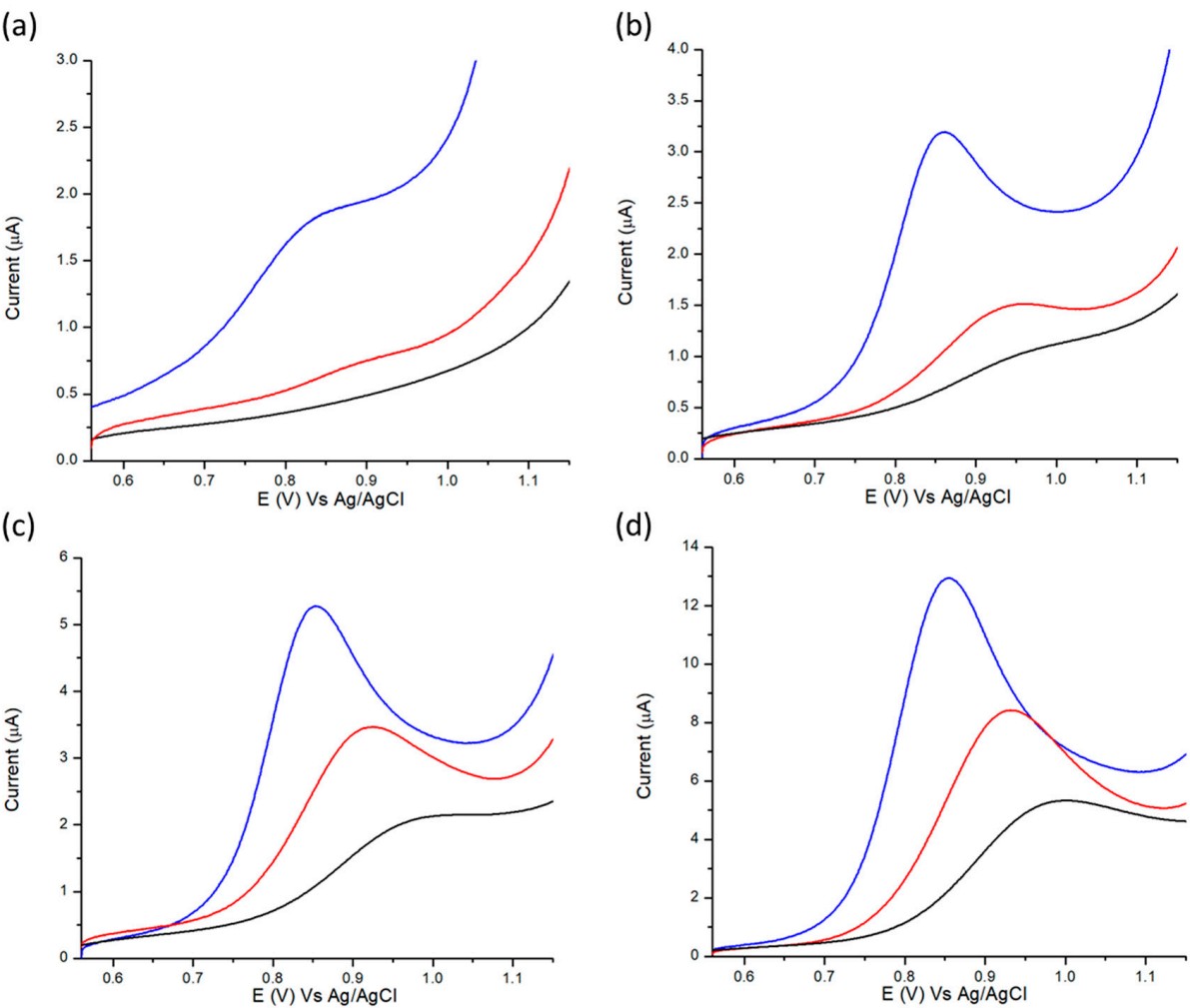

**Figure 1.** CVs of bare GCE (black line), HMT-PMBI (red line), and HMT-PMBI/NGNP (blue line) coated electrodes recorded at different concentrations of nitrites: (**a**) 1 μM, (**b**) 10 μM, (**c**) 30 μM, and (**d**) 100 μM. The supporting electrolyte was 0.1 M NaCl and scan rate was 0.1 V s$^{-1}$.

At high concentrations of H$^+$ the equilibrium is shifted towards the formation of NO$_3^-$, which cannot be further oxidized. Based on these results, pH 7 was selected as the optimum working pH for the analysis of real samples.

Figure 2a,b show the plot of the anodic peak currents vs. nitrite concentration relative to the bare GCE, HMT-PMBI, and HMT-PMBI/NGNP coated electrodes at various concentrations of nitrite (from 1 μM to 100 μM); two linear ranges were observed. It is evident that HMT-PMBI/NGNPs show much higher peak current values: at 10 μM nitrite, the peak current was ca. 20-fold higher than the bare GCE and over 4-fold higher than HMT-PMBI coated electrodes, as a result of the preconcentrating capability of the ionomer along with the higher electrical conductivity caused by the addition of NGNPs.

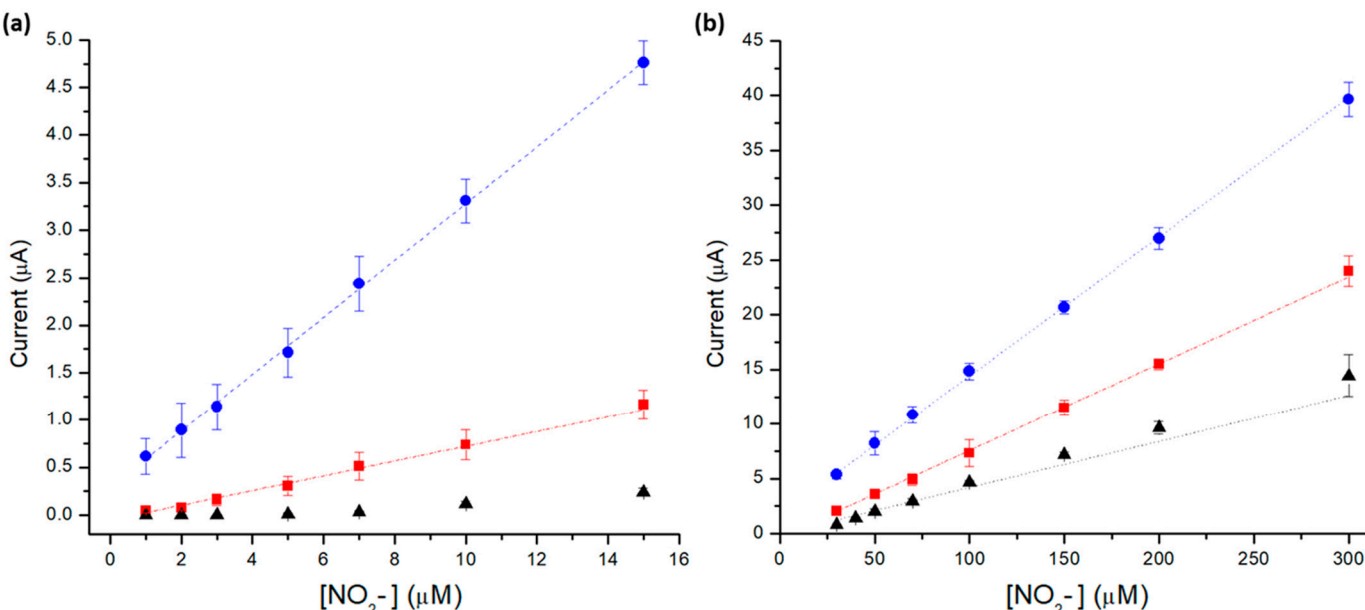

**Figure 2.** Plot of anodic peak currents vs. concentration of nitrites for bare GCE (black triangles), HMT-PMBI (red squares) and HMT-PMBI/NGNP coated electrodes (blue circles)—(**a**) from 1 μM to 15 μM; (**b**) from 30 μM to 300 μM. The supporting electrolyte was 0.1 M NaCl and scan rate was 0.1 V s$^{-1}$. Error bars were calculated from three repeat measurements.

These data highlight that HTM-PMBI and HMT-PMBI/NGNP coated electrodes are effective in preconcentrating nitrite ions. The peak current varied linearly with the concentration of nitrite ions, showing two distinct linear ranges, i.e., between 0.6 μM–15 μM and 30 μM–300 μM for the HTM-PMBI-coated electrodes and between 0.72 μM–15 μM and 30 μM–300 μM for the HMT-PMBI/NGNPs coated electrodes. The linear regression equation for the HMT-PMBI coated electrode was calculated as $I_p$ (μA) = 0.08 (μA μM$^{-1}$) (nitrites) (μM)–0.05 (μA), with the coefficient of determination R$^2$ = 0.988 (*n* = 3) and uncertainty (i.e., with a 95% confidence interval) calculated as 0.8 ± 0.008 for the slope and −0.05 ± 0.025 for the intercept. In the case of HMT-PMBI/NGNPs coated electrodes, the linear regression equation was found to be $I_p$ (μA) = 0.3 (μA μM$^{-1}$) (nitrites) (μM) + 0.29 (μA), with the coefficient of determination R$^2$ = 0.999 (*n* = 3) and uncertainty calculated as 0.3 ± 0.02 for the slope and 0.29 ± 0.14 for the intercept [50]. The calculated limits of detection (LoD) were 0.6 μM for PMBI and 0.72 μM for HMT-PMBI/NGNPs coated electrodes and were obtained from the standard deviation (S$_b$) of three blank experiments and the slope of the calibration plot using the relation LoD = 3.3 S$_b$/S [51]. It is worth mentioning that the error bars obtained from three different modified electrodes were larger in the case of the HMT-PMBI/NGNPs coated electrodes. This behavior was expected, considering that NGNPs were randomly deposited on the GCE, with a significant variation in the active site available depending on their exposure on the electrode surface. The analysis performed using CV allowed the identification of the potential of oxidation of nitrites to nitrates, as well as the evaluation of the general electrochemical properties of the composite coated films. As the HMT-PMBI/NGNP coated electrodes (0.3 μA μM$^{-1}$) showed a higher sensitivity compared to the bare GCE and the HMT-PMBI coated electrodes (0.08 μA μM$^{-1}$), from this point on we focused on the former to further investigate the performance of the HMT-PMBI/NGNPs for nitrite detection. Moreover, we used chronoamperometry, in which the potential is stepped at the value of the oxidation of nitrites to measure the resulting faradaic current as a function of the time and of the concentration of nitrites. This technique is faster than CV and facilitates the identification of potential interferences at the value of the applied potential. Figure 3 illustrates the amperometric *i–t* response obtained at a HMT-PMBI/NGNP modified electrode in the presence of various concentrations of sodium nitrite. The applied potential was 0.86 V in order to ensure the complete oxidation

of nitrites and the experiment was performed under continuous stirring, with different concentrations of nitrites added every 30 s. The *i–t* curve depicts a fast response at each addition of nitrite with an increase in the current, which rapidly reached a steady-state value. The plot of the current vs. the nitrite concentration is shown in Figure 3b.

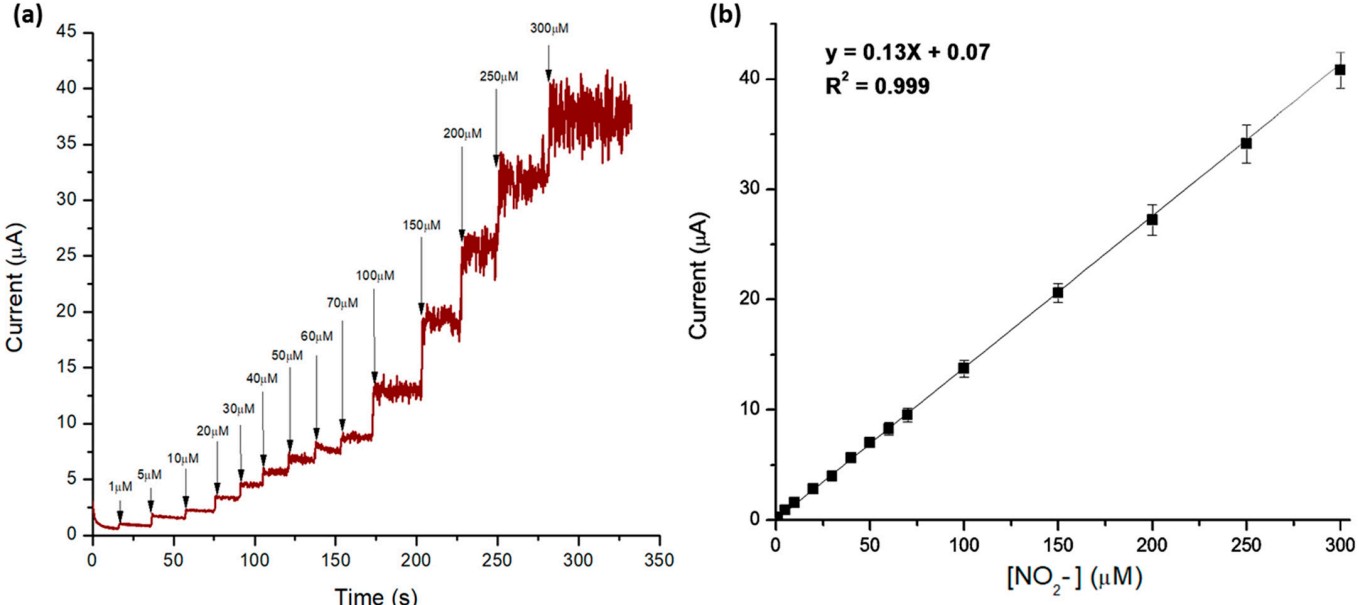

**Figure 3.** (**a**) Chronoamperometric (*i–t*) response of a HMT-PMBI/NGNP coated electrode obtained with the successive addition of nitrite from 1 μM to 300 μM. The supporting electrolyte was 0.1 M NaCl (pH 7) and applied potential was 0.86 V. (**b**) Calibration plot of the chronoamperometric (*i–t*) current vs. the nitrite concentration.

The plot shows a linear response in the range from 1 μM to 300 μM, with a regression equation expressed as $I_p$ (μA) = 0.13 (μA μM$^{-1}$) (nitrite) (μM) + 0.07 (μA) and with the uncertainty calculated as 0.13 ± 0.002 for the slope and 0.07 ± 0.02 for the intercept [50]. The limit of detection (LoD) of the HMT-PMBI/NGNPs coated electrode was calculated as 0.64 μM, obtained from the standard deviation ($S_b$) of three blank experiments and the slope of the calibration curve using the relation LoD = 3.3 $S_b/S$. An important characteristic of a chemically modified electrode is its selectivity, i.e., the ability of the ionomer to discriminate analytes that could interfere with the voltammetric detection. For this purpose, the *i–t* curves were recorded in the presence of various interferences such as nitrates, calcium, zinc, ammonium, and sulphates ions, ascorbic and citric acids; and glucose. Figure S11 shows the amperometric *i–t* response recorded in a solution containing 100 μM of nitrites after the addition of several interferences. The concentration of each interference species was in excess and up to 100-fold compared to the concentration of nitrite. The results show that in the case of HMT-PMBI/NGNP coated electrode, the response depended only on the nitrite concentration and that other analytes added to the solution did not appreciably interfere with the amperometric response. In fact, the HMT-PMBI/NGNP coating (positively charged) was not only effective in repelling analytes of the same charge, but also acted as a barrier to the diffusion of neutral species at the electrode surface. For the bare GCE, by contrast, the addition of 10 mM ascorbic acid caused significant interference, with a large increase in the current (Figure S11b). While the diffusion of sodium ascorbate was retarded at the HMT-PMBI/NGNP coated electrode at least on the timescale of the experiment, using the bare GCE this species was oxidized to its dehydroascorbic acid form. This aspect is significant in food analysis, as often ascorbate is added as an antioxidant agent along with nitrites in several foods; hence, a HMT-PMBI/NGNP coated electrode provides a sensitive platform for the detection of nitrites without interference from ascorbate.

### 3.2. Assessment of Extraction of Nitrites from Meat Products Using Microwave-Assisted Heat Treatment

In Section 2.5 we described four methods using different conditions for the extraction of nitrites using the MAH treatment. The various nitrite solutions extracted using microwaves are reacted with the reagents (NED and sulfanilamide) following the colorimetric procedure reported in the AOAC method (see Supplementary Material, S3b). In this way, the measure of the absorbance derived from the samples obtained using the MAH method can be directly compared with the samples in which the nitrite content was calculated using the standard AOAC procedure. The nitrite content of each amount of meat/volume of water ratio for both the ISO 2918 and AOAC methods were calculated using the procedure described in the Supplementary Material, S3a,b. The colorimetric measurements were applied to each microwave method and each of them was repeated three times, showing similar results, with less than 20% deviation from the mean value. Method 1 showed a noticeably lower concentration of nitrites than the other processes, suggesting that high temperatures are needed to achieve a more effective extraction of nitrite. This is not surprising, considering that both the ISO and AOAC methods required the use of water at a temperature higher than 70 °C. The concentration of nitrites follows this sequence: Method 1 < Method 4 < Method 3 < Method 2. This fact suggests that reaching boiling point and maintaining it without interruption is crucial to an effective extraction of nitrite ions from meat. Based on this evidence, Method 2 was selected as the best method for the extraction of nitrites. Then, we studied the effect of the microwave time on the extraction of nitrites. Table S12 shows the results obtained using different bacon/water ratios and the effects of the microwave time on nitrite extraction. The optimization of the time was achieved by comparing multiple colorimetric measurements. As expected, the higher the amount of water, the higher the time needed for the water to reach the boiling point. Table S13 shows the amount of $NaNO_2$ (in μg) obtained per gram of bacon and the corresponding standard deviations. The concentration of nitrites was found to be higher using lower meat quantities. It is important to note that the concentration of the solution was relative to the grams of bacon used in relation to the amount of water added to the sample. The fact that the nitrites concentration of the final solution was higher in sample C proves that the nitrites were dissolved more efficiently in the presence of higher amounts of water. This behavior is in agreement with that reported by Mohamed et al. using the AOAC method [37]. Notably, as processed meat (for example, bacon) contains nitrites along with high amounts of additional salts (nitrates, chlorides etc.) and various antioxidant agents, the use of small volumes of water in the preparation would yield to salt-saturated solutions, which would inhibit the full extraction of nitrites. The nitrite concentration found using the ISO and AOAC methods are in good agreement with sample C of the microwave method, with variance calculated as 0.02 and 0.03, respectively. Consequently, the MAH method is effective for a fast extraction of nitrites without the use of additional chemicals in the solution. It is worth mentioning that sample A was a very fast experiment for several reasons: short heating times, fast cooling times, and it was filtered quickly. Moreover, sample A was very repeatable (less than 10% of difference in three different samples), and so it could be utilized as a fast procedure for assessing the nitrite contents using a correction factor. It is worth highlighting that our MAH method requires only the heat and pressure exerted by microwaves, water as an extraction solvent without any other costly and toxic chemicals, and it is simple and fast.

### 3.3. Electrochemical Detection of Nitrites in Meat Products

In the previous section we demonstrated that microwave-assisted heating is a fast, reliable, and efficient method for extracting nitrite ions from cured meat samples. In this section we show that HMT-PMBI/NGNP coated electrodes are suitable for the electrochemical detection of nitrites in bacon. The bacon was processed using the microwave method (procedure C). A quantity of 10 mL of the final solution (500 mL) was directly placed in the electrochemical cell and analyzed. No filtration, dilution, or pretreatment steps were

required. The pH of the sample was found to be almost neutral (pH~6.7), so the samples did not require a pH adjustment. Figure 4a shows the LSVs of the solution extracted from the bacon and after addition of different concentration of nitrite. The irreversible oxidation peak of nitrite is visible (red curve) at ca. 0.86 V, with a current intensity of 0.54 µA. The intensity of the anodic peak increased linearly after the addition of sodium nitrite to the untreated sample. Considering the regression line obtained in Figure 2 ($I_p$ (µA) = 0.3 (µA µM$^{-1}$) (nitrites) (µM) + 0.29 (µA)) the calculated concentration of nitrite in the sample is 0.83 µM with the uncertainty of the method calculated as $\pm$ 0.5µM. Taking into account the volume of water used (500 mL) and the quantity of meat (5 g), it was possible to calculate the overall concentration of nitrite as 5.72 µg NaNO$_2$/g bacon with the uncertainty of the method calculated as $\pm$ 3.3 µg NaNO$_2$/g bacon. Figure 4b presents the standard addition plot corresponding to the current vs. the added nitrite concentration. The regression equation was found to be $I_p$ (µA) = 0.63 (µA µM$^{-1}$) (nitrites) (µM) + 0.56 (µA) and with the uncertainty calculated as 0.63 $\pm$ 0.03 for the slope and 0.56 $\pm$ 0.2 for the intercept. The concentration of nitrites in the bacon solution calculated using the standard addition (intercept/slope) method was 0.89 µM with the uncertainty of the method calculated as $\pm$0.2 µM. This value corresponds to 6.1 µg NaNO$_2$/g bacon with the uncertainty of the method calculated as $\pm$1.7 µg NaNO$_2$/g bacon. This result agrees well with the values obtained using the colorimetric method (see Table S1).

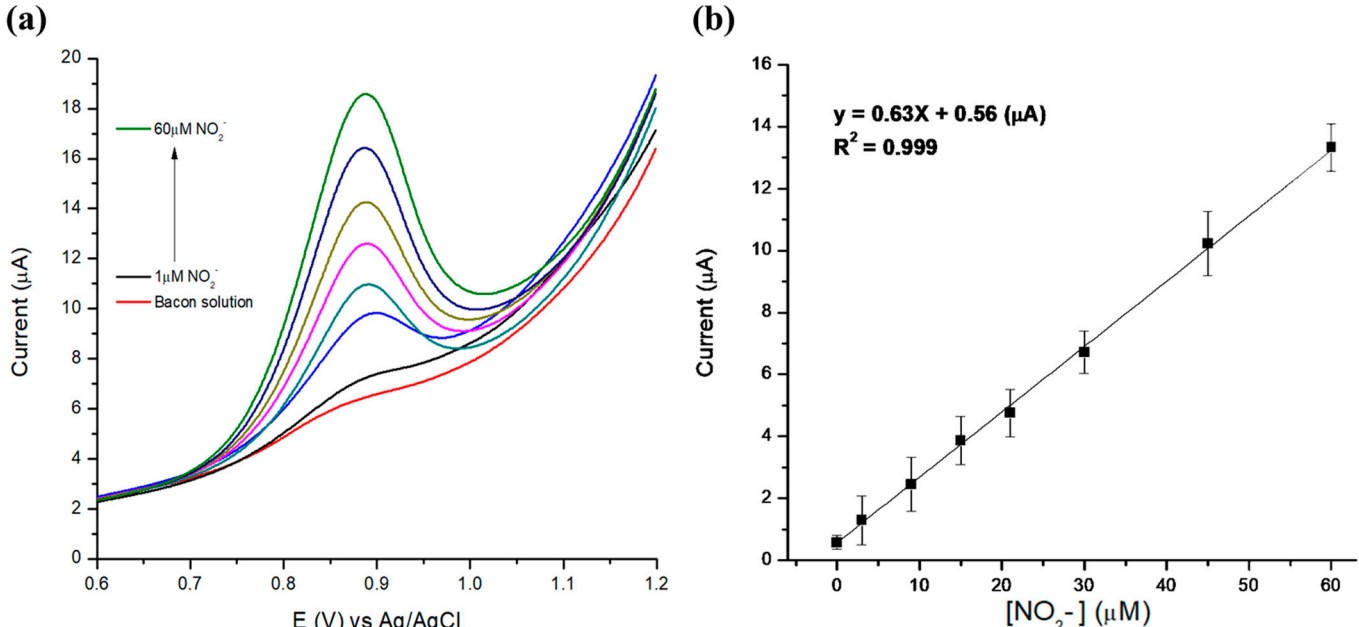

**Figure 4.** (**a**) CVs of the HMT-PMBI/NGNP coated electrode recorded in solutions containing nitrites extracted from bacon (red line) and after the addition of nitrites from 1 µM to 60 µM. (**b**) Plot of peak current vs. added concentration of sodium nitrite. The supporting electrolyte was 0.1 M NaCl and scan rate was 0.1 V s$^{-1}$. Error bars were calculated from three repeat measurements.

The electrochemical detection of nitrites in the bacon solution was also studied using chronoamperometric measurements. The solution was used without any pretreatments or dilutions, and nitrites were added to the solution during the chronoamperometric experiments. Figure 5a shows the amperometric *i–t* graph obtained after the addition of different concentrations of nitrites, whereas Figure 5b represents the calibration plot corresponding to the peak current vs. added nitrite concentration. The calculated linear regression was $I_p$ (µA) = 0.115 (µA µM$^{-1}$) (nitrites) (µM) + 0.1 (µA) with the uncertainty calculated as 0.115 $\pm$ 0.003 for the slope and 0.10 $\pm$ 0.04 for the intercept.

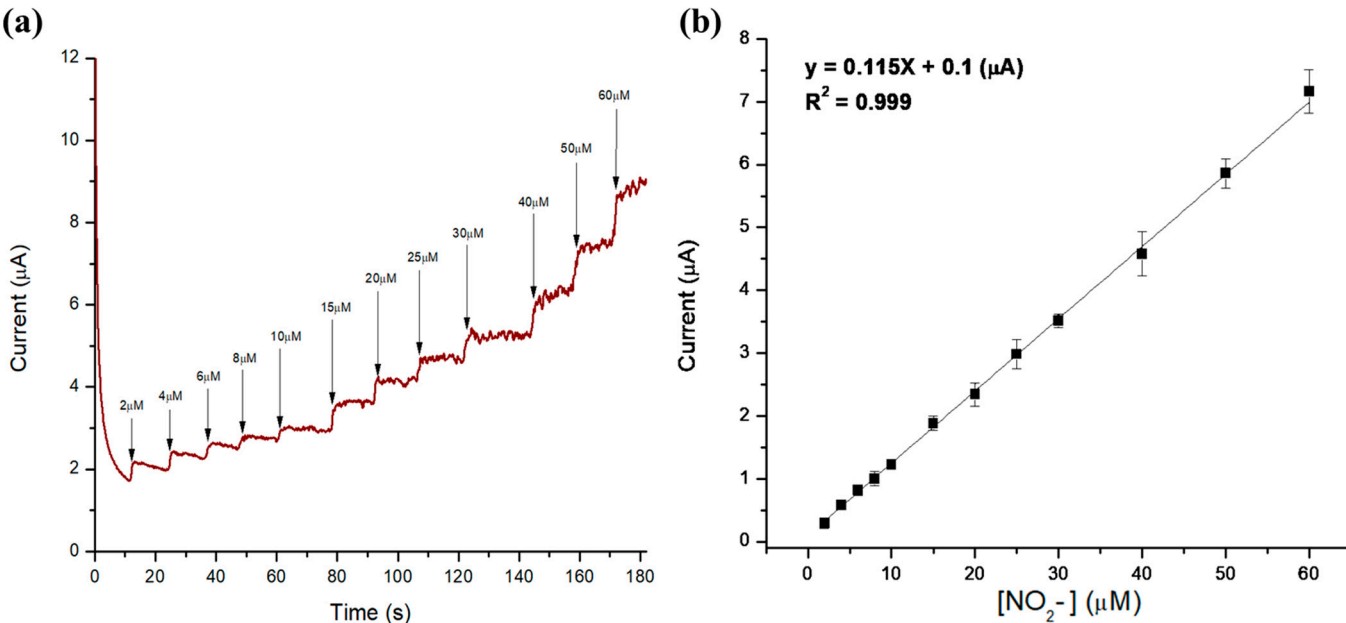

**Figure 5.** (**a**) Chronoamperometric (*i–t*) response of the PMBI/NGNPs coated electrode obtained in the solution containing nitrites extracted from bacon and after the addition of different concentrations of sodium nitrites. (**b**) Calibration plot as a function of the nitrite concentration. The applied potential was 0.86 V and error bars were calculated from three repeat measurements.

Therefore, the concentration of nitrites in the bacon solution was calculated as 0.87 μM, with the uncertainty of the method calculated as $\pm 0.3 \mu$M., which corresponds to 6 $\pm$ 2 μg NaNO$_2$/g bacon. The different concentrations of nitrites found in the commercial bacon using colorimetric and electrochemical methods are summarized in Table 1.

**Table 1.** Summary of the calculated concentrations of nitrites found in the same brand of commercial bacon using different methods. The values correspond to an average value obtained from three repeat measurements.

| Method | (Nitrite) (μM) | (NaNO$_2$) (μg/g Bacon) |
|---|---|---|
| ISO (colorimetric) | $0.804 \pm 0.02$ | $5.56 \pm 0.15$ |
| AOAC (colorimetric) | $0.814 \pm 0.03$ | $5.62 \pm 0.26$ |
| Microwave (colorimetric) | $0.777 \pm 0.06$ | $5.36 \pm 0.48$ |
| Microwave (CV curve) | $0.83 \pm 0.16$ | $5.72 \pm 0.92$ |
| Microwave (CV standard addition) | $0.89 \pm 0.08$ | $6.10 \pm 0.46$ |
| Microwave (CA standard addition) | $0.87 \pm 0.11$ | $6.00 \pm 0.64$ |

All the methods showed similar results, which highlight the suitability of HMT-PMBI/NGNP coated electrodes for the electrochemical detection of nitrites. Therefore, HMT-PMBI/NGNP coated electrodes are a good platform for the detection of nitrites without any pretreatment, and it does not require the use of toxic and expensive chemicals, such as those utilized in the colorimetric methods. The results showed good reproducibility and recovery values for the CV and chronoamperometric (*i–t*) methods, with recoveries of 109.3 and 102.7%, respectively (see Table S14). It is worth comparing the performances of HMT-PMBI/NGNPs with other modified electrodes reported in the literature (see Table S15). Even though HMT-PMBI/NGNPs do not provide lower limits of detection than other materials, we point out the simplicity of our approach, which consists in mixing nitrogen-doped graphite nanoplatelets with an anion-exchange ionomer to make a composite that is simply recast on a glassy carbon electrode. To underline the fact,

our approach does not require any treatment or dilution of the sample that typically is required when the modified electrode is too sensitive. HMT-PMBI/NGNPs along with the MAH procedure allows the determination of nitrites in a simple fashion using simple electrochemical techniques such as cyclic voltammetry and/or chroamperometry. These techniques are well suited to the analysis of nitrites in food in which the expected concentration is in the micromolar/micrograms concentration ranges. Obviously, the sensitivity can be further enhanced by using pulsed voltammetric techniques such as differential pulse voltammetry (DPV) and square wave voltammetry (SWV).

## 4. Conclusions

We have developed a novel method of extracting nitrites ions using microwave heating. This method is fast, cheap, reliable, and does not involve the use of toxic chemicals or laborious procedures of extraction compared to common standard methods currently utilized (ISO 2918 and AOAC 973.31). The detection of nitrites ions was achieved using voltammetric methods and a novel chemically modified electrode based on a HMT-PMBI/NGNP coating. The basic electrochemical properties of HMT-PMBI/NGNP coated electrodes were characterized using $IrCl_6^{2-/3-}$ as a redox probe, which led to the estimation of the apparent diffusion coefficient. HMT-PMBI/NGNP coated electrodes were tested for the determination of nitrite ions in samples of processed meat (bacon) using the microwave method of extraction. The results were compared with those obtained using standard extraction methods and detection using colorimetric methods. The results indicated that the as-prepared HMT-PMBI/NGNP coated films can detect nitrite ions with a limit of detection of 0.64 µM, a sensitivity of 0.52 µA·µM$^{-1}$ cm$^{-2}$, and a linearity range between 1 and 300 µM. The results reported here suggest that HMT-PMBI/NGNP modified electrodes are an effective material for the amperometric detection of redox active anionic species and that the microwave method is a more efficient and effective extraction method than the conventional ones. To underline this fact, NGNPs are very cheap ($0.06/g) and both NGNPs and HMT-PMBI are materials compatible with screen-printed technologies, which are widely utilized for the mass production of sensor strips.

**Supplementary Materials:** The following are available online at https://www.mdpi.com/article/10.3390/chemosensors9110325/s1.

**Author Contributions:** S.H.-A.: Conceptualization, Validation, Investigation, Writing—original draft, Formal analysis. A.T.: Investigation: synthesis of NGNPs, FESEM analysis. P.B.: Conceptualization, Writing—review & editing, Project administration, Supervision, Funding acquisition. All authors have read and agreed to the published version of the manuscript.

**Funding:** S.H.-A. gratefully acknowledges financial support from a Knowledge Economy Skills PhD Scholarship (KESS2) under the Welsh Government's European Social Fund (ESF) convergence programme for Wales and the Valleys.

**Institutional Review Board Statement:** Not applicable.

**Informed Consent Statement:** Not applicable.

**Acknowledgments:** The authors are grateful to Stephen Holdcroft (Simon Fraser University, Canada) for the generous gift of a HMT-PMBI ionomer.

**Conflicts of Interest:** The authors declare that they have no known competing financial interest or personal relationships that could have appeared to influence the work reported in this paper.

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
