# Peer review of "Efficient Microwave-Assisted Extraction of Nitrites from Cured Meat and Their Voltammetric Detection at Chemically Modified Electrodes Based on Hexamethyl-p-Terphenyl Poly(methylatedbenzimidazolium) Incorporating Nitrogen-Doped Graphite Nanoplatelets"

_chemosensors, doi:10.3390/chemosensors9110325_

Round 1

Reviewer 1 Report

In the manuscript, the authors reported an efficient microwave-assisted extraction of nitrites from cured meat and voltammetric detection via chemically modified electrodes. Actually, this chemical application of detection of nitrites in cured meat is important for food safety and deserves a more detailed investigation. Compared with the standard procedures such as ISO 2918 and the AOAC international 973.31 methods, the microwave-assisted methodology in this work displays a good consistency. As a result, I recommend this article to be published in Chemosensors before some corrections.

Comments:
1. Since the author states that the microwave-assisted methodology in detecting the nitrites is efficient, it is needed to cite some relevant works to introduce the advantages of micro-wave assisted methodology in the section of the introduction.

2. In the section on electrochemical detection of nitrites, the author should recorrect the “Error! Reference source not found.”

3. It is not sufficient for the author to cite 23 articles in this work, the author should cite more relevant works to improve this work.

Author Response

We thank the Referee for the useful comments.

Q1: Since the author states that the microwave-assisted methodology in detecting the nitrites is efficient, it is needed to cite some relevant works to introduce the advantages of micro-wave assisted methodology in the section of the introduction.

A1: We have cited relevant work on microwave-assisted heat, see now Refs. 38-44.

Q2: In the section on electrochemical detection of nitrites, the author should recorrect the “Error! Reference source not found.”

A2: We have corrected the mistake.

Q3: It is not sufficient for the author to cite 23 articles in this work, the author should cite more relevant works to improve this work.

A3: As a result of the improvement in the introduction section and other sections of the manuscript, the total number of references is now 56 with additional 25 references separately reported in the SI file.

Reviewer 2 Report

The submitted article reports interesting results of research on the new method of nitrite determination using chemically modified electrode. The results are convincing and well described.

Considering the technical aspects of the work, the figures and tables (Supplementary info) need attention:

  • The panel labels should be unified (font type, font size, style (a) or A), etc)
  • The figures’ panels should be aligned
  • The tables format should be unified

In line 322 the ‘Error! Reference source not found’ needs corrections.

Author Response

We thank the Referee for the useful comments.

Q1: Considering the technical aspects of the work, the figures and tables (Supplementary info) need attention.

The panel labels should be unified (font type, font size, style (a) or A), etc).

A1: We have unified the panel labels.

Q2: The figures’ panels should be aligned.

A2: We have aligned the panel figures.

Q3: The tables format should be unified.

A3: We have unified the tables format.

Q4: In line 322 the ‘Error! Reference source not found’ needs corrections.

A4: We have corrected the mistake.

Reviewer 3 Report

This manuscript describes an efficient way to detect nitrite in cured meat by electrochemical method. The target analyte is quite meaningful for real application. However,  the logic / stucture of this manuscript need to be readjusted, because the current form make the reader difficult to understand the key point of this work. The manuscript need to be simplyfied. Some specific comments:

  1. In the introduction part, authors describe well all the analytical methods despite electrochemical method. From page2, line 52, there was even no reference cited concerning electrochemical detection. It's quite suprising for a work using electrochemical method but without citing any electrochemical reference.
  2. In both the title and the experimental part, the microwave-assisted heating was emphasized (4 methods), but there was no detailed discussion on this part later. From my understanding, the construction of the electrode was the novelty of the work, because from the final table 1, the microwave(colorimetic) doesn't show better result compared to ISO and AOAC. I suggest the authors to simplify the experimental part, then the main target of this work will be more clear.
  3. There was a mistake in figure caption of Figure 2, red dots appeared two times.
  4. All the figures and calculations in 3.1 are in supporting information, I think this section should be move to SI. Moreover, the correlation of Dapp between IrCl62- and nitrite ions was not discussed.
  5. Page 8, line 336, the reason for two distinct linear ranges was not clearly presented.

Round 2

Reviewer 3 Report

The modified version of this manuscript is much more clear than the previous one, but  there is one minor problem: authors add ref 49-53 in the reference list, but they are not been cited in the manuscript.

Author Response

Q1: The modified version of this manuscript is much more clear than the previous one, but  there is one minor problem: authors add ref 49-53 in the reference list, but they are not been cited in the manuscript.

A1: Many thanks to the Referee to spotting a mistake. When we moved Section 3.1 of the first draft of the main text to the SI section, for some reasons EndNote kept the related references listed in the Reference section (these were the references spotted by the Referee). These references have now been deleted from the main text and moved into the SI section.